# Estimation of the demand for palliative care in a hospital in southern Peru

**Brayan Miranda-Chavez**[1,2,3], **Pablo Carrillo Barriga**[1], **Javier Flores-Cohaila**[4],
**Marco Rivarola Hidalgo**[1,2], **Alvaro Taype-Rondan** [5,6]*

1 Facultad de Ciencias de la Salud de la Universidad Privada de Tacna, Tacna, Perú, 2 Centro de Estudios e Investigación en Educación Médica y Bioética. Universidad Privada de Tacna, Tacna, Perú, 3 Servicio de Geriatría, Hospital Nacional Guillermo Almenara Irigoyen, EsSalud, Lima, Perú, 4 Escuela de Medicina Humana, Universidad Científica del Sur, Lima, 5 EviSalud, Evidencias en Salud, Lima, Perú, 6 Universidad San Ignacio de Loyola, Unidad de Investigación para la Generación y Síntesis de Evidencias en Salud, Lima, Perú

* alvaro.taype.r@gmail.com

## Abstract

### Objective

To estimate the need for palliative care in deceased patients at the Daniel Alcides Carrión Hospital (Tacna, Peru) during 2023.

### Methodology

An analytical cross-sectional study was conducted with data from adult patients who died in a Peruvian hospital in 2023. Data were collected through the review of death certificates and medical records using the EsSalud Intelligent Health Service. Two methods were used to estimate the need for palliative care: the Rosenwax method (medium estimation) and the Murtagh method (medium-high estimation).

### Results

Out of 255 deaths, 239 were analyzed. The median age was 76 years. 58.2% had two or more comorbidities, and only 3.8% had received palliative care previously. According to the Murtagh method, 82.9% of the patients required palliative care, while the Rosenwax method estimated a need of 74.1%. The most common diagnoses requiring palliative care were: neoplasms (33.5%), chronic obstructive pulmonary disease (15.1%), and renal disease (11.3%). An association was found between having two or more comorbidities and a greater need for palliative care.

### Conclusion

It was found that between 7 and 8 out of every 10 patients needed palliative care, an estimate higher than in other regions. Additionally, having two or more comorbidities was associated with the need for these services. These findings highlight the urgent need for palliative care services in Peru, suggesting that their proper implementation is crucial for improving end-of-life care.

**Data availability statement:** All relevant data are within the paper and its Supporting Information files.

**Funding:** The author(s) received no specific funding for this work.

**Competing interests:** NO authors have competing interests.

## Introduction

The burden of chronic and oncological diseases is increasing globally, potentially raising the risk of mortality and reducing the quality of life for patients. This situation underscores the critical need to alleviate the pain and suffering of individuals with these conditions [1].

Palliative care may be defined as "the active, holistic care of individuals of all ages with ser ious health-related suffering due to severe illness, especially those near the end of life" [2]. Globally, this need is estimated at 75% of deaths from chronic diseases [3]. Additionally, according to The Lancet report "Alleviating the access abyss in palliative care and pain relief—an imperative of universal health coverage," the need for palliative care among those who died in 2015 was estimated at 45% (25.5 out of 56.2 million) [4]. More recent data from the Global Atlas of Palliative Care in 2017 estimated this need at 82% [5]. In Latin America, the situation is quite disheartening, as by 2017, approximately 2 million people needed palliative care, but only 1% received these services [6].

It is important to note that various estimates exist regarding the need for palliative care, some more conservative than others [7]. Among these, the methodology proposed by Murtagh et al. is widely accepted for not underestimating the need for palliative care [7]. This approach improves upon previous estimates made by Higginson and Rosenwax by refining the criteria and providing a more detailed and accurate assessment [8,9].

In Peru, although there has been a National Palliative Care Plan for Oncological and Non-Oncological Diseases since 2021, it has not been effectively implemented to date. Additionally, up to the time of this study, there is only one work in Peru that estimates the need for palliative care in living patients with chronic diseases [10]. However, there are no studies that explore the needs based on deceased patients, nor in a large sample size.

Therefore, the objective of this study is to estimate the need for palliative care and its associated factors in patients who died at the Daniel Alcides Carrion Hospital (Tacna, Peru) during 2023.

## Methods and materials

### Study design and population

Analytical cross-sectional study. Data were collected from April to June 2024, involving adult patients (over 18 years old) who died during hospitalization across all departments at Daniel Alcides Carrion Hospital (Tacna, Peru) in 2023. Patients with incomplete medical records were excluded.

The Daniel Alcides Carrion Hospital (Tacna, Peru) is a third-level referral center that oversees the health of 120,000 insured individuals in the province of Tacna. Since January 2022, it has offered more than 30 specialties, including Internal Medicine, Oncology, Cardiology, Nephrology, and Rheumatology. It receives an average of 640 outpatient visits per day and has 110 hospitalization beds and 28 beds in the Intensive Care Unit [11].

### Procedures

The principal investigator (BMC) requested from the epidemiology department of the EsSalud Tacna Healthcare Network a list of deaths that occurred during hospitalization in 2023 at the Daniel Alcides Carrión Hospital, located in the city of Tacna. This list was provided in Excel format, with deaths numbered according to their order of occurrence and including the national identity document number to verify the information.

Subsequently, access was granted to the EsSalud Intelligent Health Service (ESSI), which allows for consultation of the patient's electronic medical record, containing all medical care

received, both in outpatient settings and during hospitalization. Access to this information was made using the national identity document number.

Two investigators (BMC and WZR) independently reviewed the electronic medical records of deceased patients and extracted the variables of interest into an Excel spreadsheet. Discrepancies identified during the comparison of the two extractions were resolved with the assistance of a third author (MHZ).

**Assessing the need for palliative care.** To estimate the need for palliative care, two methods were used. The first was the estimation method proposed by Rosenwax et al. [12], which establishes 10 diagnoses that justify the need for palliative care and proposes three levels of estimation:

- Minimum: Considers that palliative care was needed when the underlying cause of death was one of the 10 established diagnoses

- Medium: Considers that palliative care was needed when patients were hospitalized at least once during the 12 months prior to their death for a condition related to one of the 10 previously established diagnoses, with the condition being the underlying cause of hospitalization.

- Maximum: Considers that palliative care was required in all deaths except for sudden deaths, poisoning, injuries, or those related to pregnancy, childbirth, puerperium, and the perinatal period.

For this study, the medium estimation was used, as the minimum method could underestimate the need for palliative care [9,13].

The second method was the one proposed by Murtagh [12], which was developed after refining the ICD-10 codes proposed by its predecessors (the method proposed by Rosenwax [9] and the method proposed by Higginson [14], making several adjustments to incorporate emerging conditions. This method establishes various levels of estimation:

- Minimum: Based on the number of deaths from 10 specified conditions, broken down using ICD-10 codes.

- Low medium: Expands on the minimum estimation (as defined by Rosenwax) by also including cases of individuals who were admitted to the hospital in the year prior to their death with the same condition documented as the underlying cause of death. Additionally, it includes deaths where Alzheimer's, dementia, senility, or chronic renal failure are recorded as underlying causes. This estimation aims to identify specific diseases that are often underreported.

- High medium: Includes all deaths that mention any of the 10 conditions of interest described using ICD-10 codes on the death certificate, whether as an underlying or contributing cause.

- High: Similar to Rosenwax's maximum estimation, considers that palliative care was required in all deaths except for sudden deaths, poisoning, injuries, or those related to pregnancy, childbirth, postpartum, and the perinatal period.

For this study, the high medium estimation was used, as it most closely resembles the medium estimation by the Rosenwax method [12].

The choice of the methods proposed by Murtagh and Rosenwax is justified by four main pillars:

1. **Adaptability**: Their high capacity to adapt to different geographical and socioeconomic contexts was essential for considering the local factors that influence the demand for

palliative care [4]. In contrast, other methods, such as the one proposed by Gómez-Batiste, exhibit less flexibility in adjusting to various local realities [15].

2. **Data integration**: They facilitate the incorporation of data from multiple sources, which improves the accuracy of estimates and provides a more comprehensive view of patients' needs [15].

3. **Robustness and empirical validation**: Their robustness and empirical validation, demonstrated by Jeba et al., ensure the reliability of the method [16]. This surpasses other approaches, such as the one proposed by Howard et al., which do not provide long-term projections [17].

4. **Flexibility in the incorporation of factors**: The method's ability to integrate new weighting factors, such as those proposed by the Lancet Commission, allows estimates to be dynamically updated without compromising the analysis [4].

We chose not to use the Lancet Commission approach, although we recognize its relevance, for two reasons: (1) the methods proposed by Rosenwax and Murtagh are better suited for analyzing hospital deaths and specific diagnoses, which are directly applicable to our population, whereas the Lancet Commission approach is more appropriate for regional or national-level estimations; (2) the methods proposed by Rosenwax and Murtagh methods ensure greater comparability with previous studies conducted in similar contexts and significantly contribute to the literature in low- and middle-income countries.

## Other variables collected

Other variables were also included, such as age, sex, length of hospitalization, acute resuscitation plan (use of resuscitation measures in a patient with significant clinical deterioration), number of comorbidities, and variables related to the use of services and procedures in patients who may have needed palliative care.

Additionally, it was recorded whether the patient had received palliative care previously. A patient was considered to have received these services when their medical record explicitly specified the receipt of such services. This information was obtained from the medical record at the time of patient admission and was reviewed throughout all consultations during their last hospitalization.

## Ethical aspects

This study adhered to the international standards established by the Declaration of Helsinki. It was authorized by the ethics committee of the Faculty of Health Sciences at the Universidad Privada de Tacna, under the identification code: FACSA-CEI/047-03-2024. The committee waived the requirement for informed consent due to the retrospective nature of the study. The confidentiality of the extracted data was maintained.

## Statistical analysis

The analysis was performed using Stata V17. The variables were described in terms of frequencies, percentages, measures of central tendency, and dispersion. To evaluate the factors associated with the need for palliative care, Murtagh's method was used, as different studies highlight that this method provides the most inclusive estimates regarding the extent of palliative care needs in the population [7,13,15]. Poisson regression models were used to calculate prevalence ratios (PR) with their respective 95% confidence intervals (CI95%). This method was preferred over logistic regression due to the high prevalence of observed

palliative care needs. In high-prevalence contexts, logistic regression tends to overestimate odds ratios, which can result in less intuitive and less useful estimates for clinical interpretation. Therefore, Poisson regression with robust variance was chosen, as this approach provides more accurate and direct estimates of prevalence ratios, facilitating their interpretation in clinical terms. This is particularly relevant when describing the association between multiple comorbidities and the need for palliative care in populations with a high disease burden.

Variables with a p-value < 0.05 in the crude model were included in the adjusted model, using this threshold to select those with sufficient statistical evidence. It is important to note that, as of the time of review, no studies have been identified that link the estimation of palliative care needs with other variables. Therefore, a variable selection criterion based on preexisting theory was not applied.

## Results

At the Daniel Alcides Carrión Hospital, 255 deaths were recorded during the year 2023. Of these, four individuals were not evaluated due to errors in their death certificates, and 12 participants were excluded for not meeting the established inclusion criteria. Consequently, data from 239 participants were analyzed (Fig 1).

Study participants had a median age of 76 years. Of them, 54.8% were male, with a median hospitalization duration of 9 days. Additionally, 27.2% had an acute resuscitation plan. Among the participants, 58.2% had two or more comorbidities, with the most frequent being hypertension and neoplasms. Only 3.8% reported having received palliative care prior to their last hospitalization. Regarding patients who needed palliative care, 29.29% required ICU admission, with a median duration of 10 days (IQR 4 to 19). Finally, the median number of imaging tests and blood tests performed was 2 (IQR 1 to 4) and 10 (IQR 4 to 20), respectively (Table 1).

Regarding the estimation of palliative care needs, by applying the method proposed by Murtagh et al. to our population, it was found that 90.4% of the patients required palliative care. On the other hand, using the estimation proposed by Rosenwax et al., it was determined that 74.5% of the population needed palliative care. (Table 2)

When evaluating the factors associated with the need for palliative care according to Murtagh et al.'s estimation, it was found that patients with two or more comorbidities were 20% more likely to need palliative care compared to those with no comorbidities (adjusted PR: 1.20, 95% CI: 1.05 to 1.41). (Table 3)

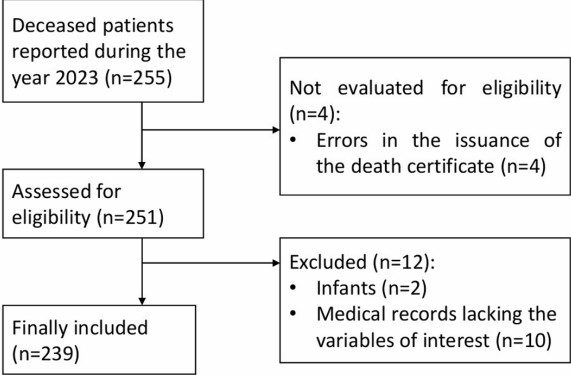

**Fig 1. Study population.**

**Table 1. Characteristics of Deceased Patients at Daniel Alcides Carrión Hospital (n = 239).**

| Variables | n (%) |
|---|---|
| Age in years* | 76 (66 to 84) |
| Age in tertiles | |
| 31 a 70 | 86 (36.0) |
| 71 a 82 | 77 (32.2) |
| 83 a 98 | 76 (31.8) |
| Male sex | 131 (54.8) |
| Duration of the last hospitalization in days * | 9 (4 to 17) |
| Acute Resuscitation Plan at Patient Admission # | 65 (27.2) |
| Number of Comorbidities | |
| None | 19 (7.9) |
| One | 81 (33.9) |
| Two or more | 139 (58.2) |
| Comorbidities^ | |
| Type 2 Diabetes Mellitus | 55 (23.0) |
| Arterial hypertension | 82 (34.3) |
| Stroke | 33 (13.8) |
| Liver Disease | 29 (12.1) |
| Neoplasm | 80 (33.5) |
| Epidemic Diseases | 32 (13.4) |
| COPD (Chronic Obstructive Pulmonary Disease) | 21 (8.8) |
| Kidney Disease | 66 (27.6) |
| Heart Disease | 32 (13.4) |
| Received Palliative Care Previously According to Medical Record | 9 (3.8) |
| Palliative care need according to Murtagh's Estimation (High medium estimation) | **216 (90.4)** |
| Number of Emergency Department Visits in the Last 6 Months for Patients with Palliative Care Need | 2 (1 to 4) |
| Number of outpatient visits in the last 6 months for patients with palliative care need | 8 (4.5 to 13) |
| Length of hospitalization in days for patients with palliative care need | 9 (4 to 17) |
| Acute resuscitation plan for patients with palliative care need | 57 (26.4) |
| ICU admission for patients with palliative care need | 62 (28.7) |
| ICU stay duration in days for patients with palliative care need | 10 (4 to 19) |
| Number of imaging tests performed during the last hospitalization for patients with palliative care need | 2 (1 to 4) |
| Number of blood tests performed during the last hospitalization for patients with palliative care need | 10 (5 to 21) |

*Median (Interquartile Range).

#Acute resuscitation plan includes having received vasopressors or cardiopulmonary resuscitation maneuvers.

^Some patients had more than one comorbidity.

## Discussion

The estimation of the need for palliative care according to Murtagh (high medium estimation) indicates that 90.4% of patients required this service. This result is similar to a 2019 study that also used the high medium estimation, conducted in England, Scotland, Northern Ireland, and Wales, reporting the need for palliative care at 90.64%, 88.96%, 90.68%, and 90.34%, respectively [16].

**Table 2. Need for Palliative Care in Deceased Patients (n = 239).**

| List of diseases by diagnosis | ICD-10 | n (%) |
|---|---|---|
| **Estimation according to Murtagh (high medium estimation)** | | **216 (90.4)** |
| Malignant neoplasm | C00-C97 | 80 (33.5) |
| Chronic obstructive pulmonary disease (COPD) | J449 | 36 (15.1) |
| Renal disease | N17, N18, N28, I12, I13 | 27 (11.3) |
| Liver disease: Liver cirrhosis | K70-K77 | 16 (6.7) |
| Respiratory disease: Pneumonia | J18 | 15 (6.3) |
| Senile dementia | F03 | 9 (3.8) |
| Cerebrovascular disease | I60-I69 | 8 (3.3) |
| Heart failure | I500-I219 | 18 (7.5) |
| Neurodegenerative disease: Parkinson's disease | G20 | 3 (1.3) |
| Respiratory disease: Respiratory failure | J96 | 2 (0.8) |
| HIV/AIDS | B24 | 2 (0.8) |
| **Estimation according to Rosenwax (medium estimation)** | | **178 (74.5)** |
| Neoplasm | C00-D48 | 80 (33.5) |
| Chronic obstructive pulmonary disease (COPD) | J40, J410, J411, J418, J42, J430-449 | 36 (15.1) |
| Renal failure | N10, N11, N18, N120, N131, N132 | 27 (11.3) |
| Liver failure | K704, K711, K721, K729 | 16 (6.7) |
| Heart failure | I110, I119, I500, I501, I509, I130, I132 | 14 (5.9) |
| Parkinson's disease | G20 | 3 (1.3) |
| HIV/AIDS | B24 | 2 (0.8) |

As expected, research using the minimum estimation according to Murtagh has found lower frequencies of palliative care need. A study in Cyprus [17] and Ireland [18] reported a need for palliative care of 47.89% and 80%, respectively. On the other hand, a study based on the minimum estimation of Murtagh, which included 12 countries (Belgium, Czech Republic, France, Hungary, Italy, Spain (Andalusia), Canada, the United States, South Korea, Mexico, and New Zealand) between 2007 and 2010, found that Mexico had the lowest need for palliative care at 56%, while Hungary had the highest need at 83% [7]. Additionally, two future projections based on Murtagh's **minimum** estimation were conducted: one in Japan for the year 2040, predicting a need of 43%, and another in Malaysia for 2030, estimating a need of 71% [19]. Finally, a study using the **maximum** estimation in Germany reported the need for palliative care at 80% [13].

On the other hand, our research using Rosenwax's method with a medium estimation shows that 74.1% of the patients who died required palliative care. Although we could not find studies that used this estimation, a 2015 study conducted in a hospital in Mallorca, Spain, which used the maximum estimation, determined a need for palliative care at 88.8% [3]. In contrast, the National Palliative Care Plan in Peru, using data from the Office of General Information Technology (OGTI) and employing Rosenwax's maximum estimation, reported that 60% of all patients who died in 2016 needed palliative care [20].

As observed, there is variability between the estimations used and the results obtained. This variability is also expected due to differences in the type of hospital (general hospitals vs. specialized hospitals), the population pyramid of the patient population, comorbidities, and other characteristics [10,21,22]. This highlights the need for assessing the need for palliative care and its evolution over time within each context to make better decisions.

It is important to highlight a significant variation in our study regarding the estimations of Murtagh (90.4%) and Rosenwax (74.5%). This finding is consistent with two previous studies conducted between 2010 [7] and 2013 [13], which reported palliative care estimates according

**Table 3. Factors Associated with the Need for Palliative Care in the Studied Population (n = 239).**

| Variables | No palliative care needed n (%) | Palliative care needed n (%) | Crude PR (95% CI)* | Adjusted PR (95% CI) |
|---|---|---|---|---|
| Age in years | 73 (36 to 97) | 76 (31 to 98) | 1.00 (0.99 to 1.00) | 1.00 (0.99 to 1.00) |
| Sex | | | | |
| Female | 14 (13.0) | 94 (87.0) | REF | REF |
| Male | 9 (6.9) | 122 (93.1) | 1.03 (0.99 to 1.08) | 1.04 (1.00 to 1.08) |
| Length of hospitalization | 7 (1 a 45) | 9 (1 a 88) | 1.00 (0.99 to 1.00) | |
| Acute resuscitation plan | | | | |
| Yes | 9 (12.7) | 62 (87.3) | REF | |
| No | 14 (8.4) | 152 (91.6) | 1.02 (0.97 to 1.07) | |
| Number of comorbidities | | | | |
| None | 9 (47.4) | 10 (52.6) | REF | REF |
| One | 5 (6.2) | 76 (93.8) | 1.27 (1.09 to 1.48) | 1.22 (1.06 to 1.41) |
| Two or more | 9 (6.5) | 130 (93.5) | 1.27 (1.09 to 1.47) | 1.21 (1.05 to 1.39) |
| Emergency department visits in the last 6 months* | | | | |
| None | 10 (20.8) | 38 (79.2) | REF | REF |
| One | 5 (8.8) | 52 (91.2) | 1.07 (0.99 to 1.15) | 1.03 (0.97 to 1.10) |
| Two or more | 8 (6.0) | 126 (94.0) | 1.08 (1.01 to 1.16) | 1.03 (0.98 to 1.09) |
| Outpatient visits in the last 6 months* | | | | |
| None | 9 (34.6) | 17 (65.4) | REF | REF |
| One | 1 (7.7) | 12 (92.3) | 1.16 (1.02 to 1.33) | 1.11 (0.97 to 1.25) |
| Two or more | 13 (6.5) | 186 (93.5) | 1.17 (1.05 to 1.31) | 1.10 (0.99 to 1.22) |
| Last hospitalization | | | | |
| ICU admission | | | | |
| Yes | 9 (12.7) | 62 (87.3) | 0.98 (0.93 to 1.02) | |
| No | 14 (8.4) | 152 (91.6) | REF | |
| ICU stay duration in days | 3 (1 to 45) | 10 (1 to 146) | 1.00 (0.99 to 1.00) | |

REF: Reference value.

*Simple Poisson regression with robust variance.

**Multiple Poisson regression with robust variance; a model was generated with the variables: number of comorbidities, emergency visits in the last 6 months, and outpatient visits in the last 6 months.

to Murtagh at 74% and 78%, and according to Rosenwax at 38% and 40.7%, respectively. The difference between one estimation and another lies in the diagnoses included in each method. Murtagh considers, in addition to the diagnoses shared with Rosenwax, additional conditions such as breast, colorectal, lung, and prostate cancers, as well as cerebrovascular diseases, chronic renal insufficiency, liver diseases, and respiratory diseases. Therefore, the discrepancy in percentages may reflect the different definitions and criteria used in each method to assess the need for palliative care, highlighting the importance of considering multiple approaches to obtain a comprehensive view of palliative care needs in different populations and contexts. It is important to mention that the method proposed by Murtagh is based on the **cause of death** for estimation, whereas the Rosenwax method uses the **cause of hospital admission**. These methodological differences can lead to divergent estimations, which may underestimate or overestimate the need for palliative care depending on the available data and the context in which they are applied.

Our results underscore the need to establish palliative care services in the evaluated hospital. Although there is no specific threshold that determines the need to establish palliative care services, these estimations are a key resource for the formulation of health policies that

address the lack of this service. It is noteworthy that the evaluated hospital does not have a palliative care unit and, in general, in Peru, as of 2023, there are only 19 palliative care units, reflecting a lack of culture around this service [23].

In the Latin American context, there is currently limited literature estimating the need for palliative care at both hospital and national levels. In Chile, two studies have been published that assess the demand for palliative care based on the proportion of deceased patients requiring such services, taking into account the burden of severe health-related suffering. This methodology aligns with the recommendations of the Lancet Commission report [4]. The first study, which applied this methodology to patients who died between 2018 and 2020, found that 65% required palliative care [24]. The second study, encompassing a broader population (deaths from 1997 to 2019), reported that the need for palliative care was 44.5% in 1997 and increased to 48% in 2019 [15]. Additionally, it projects that this need will rise from 58,000 patients in 2021 to 105,000 by 2050. In Brazil, a study analyzing death records of cancer patients from 2008 to 2014, utilizing the World Health Organization's model to estimate palliative care needs, found that 80% of terminal cancer patients required these services [25].

Regarding the associated factors, it was reported that having two or more comorbidities was associated with the need for palliative care. This suggests that patients with more complex health conditions are likely to require these services; however, future studies could determine whether this and other factors might be useful for predicting the need for palliative care in the Peruvian context [26,27]. In contrast to our results, studies conducted in other contexts found a higher need for palliative care in older individuals and in women [7,28,29]. These differences may be due to variations in disease burden among subpopulations, which should be studied in greater depth [30].

We found that, during their last hospitalization, 29.3% of the deceased were admitted to the ICU, and 27.8% had an acute resuscitation plan that involved receiving cardiopulmonary resuscitation or the administration of vasopressors. It would be of interest for future studies to assess whether these procedures were beneficial. In this regard, a systematic review, which included studies from 1995 to 2015, concluded that 30% of patients nearing death underwent unnecessary procedures [31]. Finally, this scenario could be attributed to a combination of the culture of 'doing everything possible,' the fear of death, and a false sense of hope from healthcare personnel or the patient's family members. This leads to a series of repercussions, both financially for the healthcare system and, most importantly, in the disregard for the patient's dignity and quality of life. It is important to mention that, although 36% of the patients were between 31 and 70 years old and 8% had no comorbidities, the analysis does not fully capture the clinical reasons that justified their admission to the ICU. In the case of young patients without comorbidities, these decisions are often considered appropriate due to their higher likelihood of recovery.

## Limitations and strengths

It is important to consider certain limitations of the study. This was conducted in a hospital that provides services to a specific population within Peru, with its own characteristics and trends. Additionally, there is a possibility that the data collected from death records is underreported, especially regarding neurological diseases such as dementia and Parkinson's. Likewise, our estimation, being based on the primary diagnosis of the cause of death, could underestimate the need for palliative care, as comorbidities, which could increase the demand for these services, are not fully considered. Furthermore, this study only assessed palliative care needs among hospitalized patients, which is a significant limitation as a large proportion of individuals requiring palliative care die in the community. The method used (Murtagh) should account for all deaths mentioning any of the 10 conditions of interest, not just those occurring in hospitals.

However, this is one of the first studies to evaluate the need for palliative care in Peru, and it is one of the few studies in Latin America that uses two estimations for this purpose.

## Conclusion

We found that 90.4% and 74.5% of hospital deaths had a need for palliative care, according to the Murtagh and Rosenwax estimations, respectively. The diagnoses that most contributed to these estimations were malignant neoplasms, chronic obstructive pulmonary disease, and renal and hepatic insufficiencies. The need for palliative care was associated with having more comorbidities and having had an outpatient visit in the last six months.

## Acknowledgment

There are no acknowledgements mentioned in this section.

## Author contributions

**Conceptualization:** Brayan Miranda-Chavez, Pablo Carrillo Barriga, Javier Flores-Cohaila, Marco Rivarola Hidalgo, Alvaro Taype-Rondan.

**Data curation:** Brayan Miranda-Chavez.

**Formal analysis:** Brayan Miranda-Chavez.

**Investigation:** Brayan Miranda-Chavez, Javier Flores-Cohaila, Alvaro Taype-Rondan.

**Methodology:** Brayan Miranda-Chavez, Alvaro Taype-Rondan.

**Writing – original draft:** Brayan Miranda-Chavez, Pablo Carrillo Barriga, Javier Flores-Cohaila, Marco Rivarola Hidalgo, Alvaro Taype-Rondan.

**Writing – review & editing:** Brayan Miranda-Chavez, Pablo Carrillo Barriga, Javier Flores-Cohaila, Marco Rivarola Hidalgo, Alvaro Taype-Rondan.

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
