## [Decision Letter · Decision Letter 0]

18 Oct 2024

Dear Dr. Taype-Rondan,

Thank you for submitting your manuscript to PLOS ONE. After careful consideration, we feel that it has merit but does not fully meet PLOS ONE’s publication criteria as it currently stands. Therefore, we invite you to submit a revised version of the manuscript that addresses the points raised during the review process.

Please submit your revised manuscript by Dec 02 2024 11:59PM. If you will need more time than this to complete your revisions, please reply to this message or contact the journal office at plosone@plos.org . A rebuttal letter that responds to each point raised by the academic editor and reviewer(s). You should upload this letter as a separate file labeled 'Response to Reviewers'.A marked-up copy of your manuscript that highlights changes made to the original version. You should upload this as a separate file labeled 'Revised Manuscript with Track Changes'.An unmarked version of your revised paper without tracked changes. You should upload this as a separate file labeled 'Manuscript'.

We look forward to receiving your revised manuscript.

Kind regards,

Luca Valera

Academic Editor

PLOS ONE

Journal Requirements:

Additional Editor Comments :

Please read carefully the comments and reviews made by the two referees and respond to them.

Reviewers' comments:

Reviewer's Responses to Questions

**Comments to the Author**

1. Is the manuscript technically sound, and do the data support the conclusions?

Reviewer #1: Partly

Reviewer #2: Partly

2. Has the statistical analysis been performed appropriately and rigorously?

Reviewer #1: Yes

Reviewer #2: No

3. Have the authors made all data underlying the findings in their manuscript fully available?

Reviewer #1: Yes

Reviewer #2: Yes

4. Is the manuscript presented in an intelligible fashion and written in standard English?

Reviewer #1: No

Reviewer #2: Yes

Reviewer #1: Thank you for study on estimating the demand for palliative care in a hospital in southern Peru. Below are some observations that I believe could strengthen your manuscript.

Presentation: The manuscript contains some inconsistencies that could be addressed to enhance its overall presentation, improving readability. I suggest a thorough revision to correct formatting and organizational issues.

Methodological Justification: While you have employed two well-recognized methods (Rosenwax and Murtagh) to estimate the need for palliative care, it would be valuable to include a more detailed discussion on why you selected these approaches over others reported in the literature. There are alternative methodologies that could have been considered, and addressing this would add robustness to your approach.

I suggest to read the following papers:

Etkind SN, Bone AE, Gomes B, Lovell N, Evans CJ, Higginson IJ, et al. How many people will need palliative care in 2040? Past trends, future projec-tions and implications for services. BMC Med. 2017;15(1):102.

Knaul FM, Farmer PE, Krakauer EL, De Lima L, Bhadelia A, Jiang Kwete X, et al. Alleviating the access abyss in palliative care and pain relief-an imperative of universal health coverage: the Lancet Commission report. Lancet (London England). 2018;391(10128):1391–454

Gómez-Batiste X, Martínez-Muñoz M, Blay C, Espinosa J, Contel JC, Ledesma A. Identifying needs and improving palliative care of chronically ill patients: a community-oriented, population-based, public-health approach. Curr Opin Support Palliat Care. 2012;6(3):371–8

Jeba J, Taylor C, O’Donnell V. Projecting palliative and end-of-life care needs in Central Lancashire up to 2040: an integrated palliative care and public health approach. Public Health. 2021;195:145–51

Howard M, Hafid A, Isenberg SR, Hsu AT, Scott M, Conen K, et al. Intensity of outpatient physician care in the last year of life: a population-based retrospective descriptive study. CMAJ Open. 2021;9(2):E613-E22. 47. Kaur S, Kaur H, Komal K, Kaur P, Kaur D, Jariyal VL, et al. Need of pallia-tive care services in rural area of Northern India. Indian J Palliat Care. 2020;26(4):528–30

Updated Weights: The Lancet Commission has recently reported new weighting factors that may differ from those you have used. I believe a discussion about these differences and their potential impact on your results would enrich your analysis.

Prevalent Population: It would be interesting to complement your methodology by considering the prevalent population living with the disease and eligible for palliative care, rather than relying solely on death records. This could provide a more comprehensive view of the potential demand.

The use of Poisson regression with robust variance to calculate prevalence ratios (PR) is appropriate for the data, given the high prevalence of palliative care needs in the population. However, it would be beneficial to provide a more detailed justification for choosing this method over alternatives such as logistic regression, which is commonly used in prevalence studies. Additionally, while the adjustment for multiple variables in the multivariable model is commendable, a more thorough discussion regarding the selection of these variables would strengthen the analysis. Specifically, explaining why certain variables were chosen a priori and included in the adjusted model only if they met the p-value threshold of <0.05 in the crude model would provide greater clarity.

Specific Corrections:

Figure 1: This figure is in Spanish. Since the publication is in English, I suggest translating it to maintain consistency throughout the document.

Table 3: Regarding Table 3, could you clarify what "REF" refers to? A brief explanation would help readers understand this more clearly.

Regional Comparison: I suggest comparing your estimates with studies from other regions. For instance, Chile has two studies analyzing the demand and need for palliative care, which could serve as useful references for comparison.

Armijo, N., Abbot, T., Espinoza, M., Neculhueque, X., Balmaceda, C., 2023. Estimation of the demand for palliative care in non-oncologic patients in Chile. BMC Palliative Care 22.. https://doi.org/10.1186/s12904-022-01122-z

Leniz, J., Domínguez, A., Bone, A.E., Etkind, S., Perez-Cruz, P.E., Sleeman, K.E., 2024. Past trends and future projections of palliative care needs in Chile: analysis of routinely available death registry and population data. BMC Medicine 22.. https://doi.org/10.1186/s12916-024-03570-1

Reviewer #2: Thank you for the opportunity to review this manuscript. Overall, I believe it has some interesting findings. However, there are several aspects of the methods that need clarification. Results have several limitations as well, as the data comes from a very specific population in Peru and therefore they are non-generalizable. Nevertheless, there are very few studies reporting palliative care needs in Latin America and therefore I think it might be a contribution to the field. I am including some specific comments in the attached document.

**Do you want your identity to be public for this peer review?** For information about this choice, including consent withdrawal, please see our Privacy Policy

Reviewer #1: No

Reviewer #2: **Yes: ** Javiera Léniz

---

## [Author Response · Author response to Decision Letter 1]

17 Dec 2024

Dear Editor,

I hereby submit for your consideration the revised version of the manuscript, now titled "Estimation of the demand for palliative care in a hospital in southern Peru," for possible publication in your prestigious journal.

All co-authors have thoroughly reviewed the suggestions, comments, and annotations provided on the previous version of the manuscript, for which we received feedback on October 18 of this year. We greatly appreciate the time spent on your review and have addressed each suggestion as follows:

Reviewer #1:

1. Presentation: The manuscript contains some inconsistencies that could be addressed to enhance its overall presentation, improving readability. I suggest a thorough revision to correct formatting and organizational issues

R: Dear Reviewer, we deeply appreciate the time you have dedicated to thoroughly reading and reviewing our work. We are pleased to have your valuable observations, aimed at improving this research. Although the banner of palliative care waves fiercely around the world, in Peru this topic remains largely overlooked. Therefore, your expertise in reviewing this content is invaluable to us. Additionally, we would like to highlight that the structure of the article has been adapted in accordance with the editorial guidelines.

2. Methodological Justification: While you have employed two well-recognized methods (Rosenwax and Murtagh) to estimate the need for palliative care, it would be valuable to include a more detailed discussion on why you selected these approaches over others reported in the literature. There are alternative methodologies that could have been considered, and addressing this would add robustness to your approach.

R: Dear reviewer, we sincerely appreciate this brilliant observation. We have attached the added paragraph that explains the reasons behind our choice of these methods:

<<The choice of the methods proposed by Murtagh and Rosenwax is justified by four main pillars:

1. Adaptability: Their high capacity to adapt to different geographical and socioeconomic contexts was essential for considering the local factors that influence the demand for palliative care (4). In contrast, other methods, such as the one proposed by Gómez-Batiste, exhibit less flexibility in adjusting to various local realities (15).

2. Data integration: They facilitate the incorporation of data from multiple sources, which improves the accuracy of estimates and provides a more comprehensive view of patients' needs (15).

3. Robustness and empirical validation: Their robustness and empirical validation, demonstrated by Jeba et al., ensure the reliability of the method (16). This surpasses other approaches, such as the one proposed by Howard et al., which do not provide long-term projections (17).

4. Flexibility in the incorporation of factors: The method's ability to integrate new weighting factors, such as those proposed by the Lancet Commission, allows estimates to be dynamically updated without compromising the analysis (4). >>

3. Prevalent Population: It would be interesting to complement your methodology by considering the prevalent population living with the disease and eligible for palliative care, rather than relying solely on death records. This could provide a more comprehensive view of the potential demand.

R: Dear reviewer, thank you for your valuable observation. However, it is important to note that the hospital where this study was conducted does not have a systematic registry of the prevalence of chronic diseases. This is due to structural limitations within the healthcare system, such as a lack of resources to implement epidemiological surveillance systems and longitudinal data registries. This deficiency complicates health service planning and the accurate assessment of the potential demand for palliative care.

4. The use of Poisson regression with robust variance to calculate prevalence ratios (PR) is appropriate for the data, given the high prevalence of palliative care needs in the population. However, it would be beneficial to provide a more detailed justification for choosing this method over alternatives such as logistic regression, which is commonly used in prevalence studies. Additionally, while the adjustment for multiple variables in the multivariable model is commendable, a more thorough discussion regarding the selection of these variables would strengthen the analysis. Specifically, explaining why certain variables were chosen a priori and included in the adjusted model only if they met the p-value threshold of <0.05 in the crude model would provide greater clarity.

R: I appreciate your observation regarding the use of logistic regression. It is true that, in our study, the results show a high prevalence of palliative care needs, reaching up to 90.4% according to the estimates from Murtagh's method. In contexts of high prevalence, logistic regression tends to overestimate odds ratios, which can result in less intuitive and less useful estimates for clinical interpretation.

For this reason, we opted to use Poisson regression with robust variance. This approach provides more precise and direct estimates of prevalence ratios, which are easier to interpret in clinical terms. This is especially relevant when describing the association between multiple comorbidities and the need for palliative care in populations with a high disease burden.

A threshold of p < 0.05 was established to filter the variables that presented sufficient statistical evidence to be included in the adjusted model. It is worth mentioning that, up to the time of the review, no studies have been identified that associate the estimation of the need for palliative care with other variables. Therefore, the criterion of selecting variables based on a preexisting theory was not employed. The statistical analysis section was adapted in the following manner:

<< The analysis was performed using Stata V17. The variables were described in terms of frequencies, percentages, measures of central tendency, and dispersion. To evaluate the factors associated with the need for palliative care, Murtagh’s method was used, as different studies highlight that this method provides the most inclusive estimates regarding the extent of palliative care needs in the population (7,13,15). Poisson regression models were used to calculate prevalence ratios (PR) with their respective 95% confidence intervals (CI95%). This method was preferred over logistic regression due to the high prevalence of observed palliative care needs. In high-prevalence contexts, logistic regression tends to overestimate odds ratios, which can result in less intuitive and less useful estimates for clinical interpretation. Therefore, Poisson regression with robust variance was chosen, as this approach provides more accurate and direct estimates of prevalence ratios, facilitating their interpretation in clinical terms. This is particularly relevant when describing the association between multiple comorbidities and the need for palliative care in populations with a high disease burden.

Variables with a p-value < 0.05 in the crude model were included in the adjusted model, using this threshold to select those with sufficient statistical evidence. It is important to note that, as of the time of review, no studies have been identified that link the estimation of palliative care needs with other variables. Therefore, a variable selection criterion based on preexisting theory was not applied.>>

5. Figure 1: This figure is in Spanish. Since the publication is in English, I suggest translating it to maintain consistency throughout the document.

Table 3: Regarding Table 3, could you clarify what "REF" refers to? A brief explanation would help readers understand this more clearly.

R: Thank you for your observation. The translation of Figure 1 has been completed. Additionally, it has been specified in the table footnote that the term "REF" stands for Reference Value.

6. Regional Comparison: I suggest comparing your estimates with studies from other regions. For instance, Chile has two studies analyzing the demand and need for palliative care, which could serve as useful references for comparison.

R: Thank you for your contribution. We have conducted the search based on your suggestion and have generated a new paragraph regarding the comparison in the regional context, in this case, Latin America. Below is the paragraph:

<< In the Latin American context, there is currently limited literature estimating the need for palliative care at both hospital and national levels. In Chile, two studies have been published that assess the demand for palliative care based on the proportion of deceased patients requiring such services, taking into account the burden of severe health-related suffering. This methodology aligns with the recommendations of the Lancet Commission report (4). The first study, which applied this methodology to patients who died between 2018 and 2020, found that 65% required palliative care (18). The second study, encompassing a broader population (deaths from 1997 to 2019), reported that the need for palliative care was 44.5% in 1997 and increased to 48% in 2019 (19). Additionally, it projects that this need will rise from 58,000 patients in 2021 to 105,000 by 2050. In Brazil, a study analyzing death records of cancer patients from 2008 to 2014, utilizing the World Health Organization’s model to estimate palliative care needs, found that 80% of terminal cancer patients required these services (20). >>

Reviewer #2:

Dear reviewer, your observations have been incredibly enlightening and have undoubtedly strengthened this work, which, despite its limitations, aims to broaden the perspective on palliative care in the country.

1. Introduction

Your aim only mentions the estimation of palliative care needs but not the analysis of

associated factors. This should be included in the aim.

R: The objective has been modified as evidenced in the following sentence:

<< Therefore, the objective of this study is to estimate the need for palliative care and its associated factors in patients who died at the Daniel Alcides Carrion Hospital (Tacna, Peru) during 2023.>>

2. Methods

The data source, extraction procedures, and study period are not clearly explained and more detail is needed. What was the population of the study? Did you include all patients who died in the Daniel Alcides Carrión Hospital in 2023? But you said data was collected from April to June 2024. It is not clear what that means. Is that information regarding hospital admissions? Mortality records? Why only April to June if all decedents in 2023 were included? Later in that section, it said “The principal investigator (BMC) requested the death records for 2023 from the epidemiology department of the EsSalud Tacna Healthcare Network.”. So it is not clear if you used information from mortality records, hospital records, or both and why the years differ. You mentioned two investigators independently extracted the variables of interest. Is that regarding hospital records? It would be useful to describe what information was extracted. If it was independently extracted, then what happened if discrepancies were found? You also said that was done anonymously. How could that have been done if you were looking at hospital records? How information from death records were linked to hospital records then? You mentioned “incomplete medical records were excluded”. Why medical records were incomplete to the point you had to exclude them? What information was incomplete in medical records? Which proportion of all the records was that? So, in summary, you need to clarify the population, study period, what information was extracted and used from these two different sources and how was it linked.

R: << Analytical cross-sectional study. Data were collected from death records of patients who died during hospitalization across all departments at the Daniel Alcides Carrión Hospital (Tacna, Peru) in 2023. Adult patients (over 18 years old) were included, and those whose identification documents did not match the death records were excluded.

The principal investigator (BMC) requested from the epidemiology department of the EsSalud Tacna Healthcare Network a list of deaths that occurred during hospitalization in 2023 at the Daniel Alcides Carrión Hospital, located in the city of Tacna. This list was provided in Excel format, with deaths numbered according to their order of occurrence and including the national identity document number to verify the information. Subsequently, access was granted to the EsSalud Intelligent Health Service (ESSI), which allows for consultation of the patient’s electronic medical record, containing all medical care received, both in outpatient settings and during hospitalization. Access to this information was made using the national identity document number.

Two investigators (BMC and WZR) independently reviewed the electronic medical records of deceased patients and extracted the variables of interest into an Excel spreadsheet. Discrepancies identified during the comparison of the two extractions were resolved with the assistance of a third author (MHZ). >>

In the “assessing the need for palliative care” section you describe the Rosenwax method and

mention “related to one of the 10 previously established diagnoses”. You should describe what

are those 10 diagnoses and if you used ICD10 codes. A table with the conditions and ICD10

codes used in the two methods would be helpful. It is also not clear if you used only the last

hospital admission or all admissions in the last 12 months to derive the number of people with

palliative care needs from Murtagh’s method.

R: Supplementary tables related to the Rosenwax and Murtagh methods have been added as supplementary material, as shown below:

Supplementary Material:

AGREGAR TABLA AQUÍ (PABLO)

All hospital admissions made in the last 12 months were considered to evaluate whether the cause of death had also been, at any time, the cause of a previous hospitalization within that time period.

Which statistical software was used?

R: <<The analysis was performed using Stata V17.>>

Which variables did you include in the multivariate model? And how did you decide which one

to include?

R: <<Variables with a p-value < 0.05 in the crude model were included in the adjusted model, using this threshold to select those with sufficient statistical evidence. It is important to note that, as of the time of review, no studies have been identified that link the estimation of palliative care needs with other variables. Therefore, a variable selection criterion based on preexisting theory was not applied. >>

How comorbidities were extracted and classified? In table 1 you describe 19 participants

without comorbidities. So how did you decide when a condition was a comorbidity and when

the main condition? Did you consider all chronic conditions or just a group of them?

R: Regarding comorbidities, all chronic diseases that the patient had were included in this group. On the other hand, the group of 19 individuals "Without comorbidities" refers to those who may have died due to traffic accidents, poisoning, or an acute illness. Additionally, the primary condition was defined as the disease that was the direct cause of the patient's death, as recorded in the electronic medical record.

Results

Your figure 1 is in Spanish

R: The figure has been translated into English.

PEGAR FIGURA

How was the Length of hospitalization in days for patients with palliative care needs calculated? Did you use all admissions in the last year of life or just the last one? And what is the difference between this variable and the “Hospitalization time in days”?

There has been an error on our part when drafting this section. We considered the variable “Duration of hospitalization” as the days that patients with palliative care needs were hospitalized during their last hospitalization. This has been corrected in the following paragraph:

<< Other v

---

## [Decision Letter · Decision Letter 1]

22 Jan 2025

Dear Dr. Taype-Rondan,

Thank you for submitting your manuscript to PLOS ONE. After careful consideration, we feel that it has merit but does not fully meet PLOS ONE’s publication criteria as it currently stands. Therefore, we invite you to submit a revised version of the manuscript that addresses the points raised during the review process.

Please submit your revised manuscript by Mar 08 2025 11:59PM. If you will need more time than this to complete your revisions, please reply to this message or contact the journal office at plosone@plos.org . A rebuttal letter that responds to each point raised by the academic editor and reviewer(s). You should upload this letter as a separate file labeled 'Response to Reviewers'.A marked-up copy of your manuscript that highlights changes made to the original version. You should upload this as a separate file labeled 'Revised Manuscript with Track Changes'.An unmarked version of your revised paper without tracked changes. You should upload this as a separate file labeled 'Manuscript'.

We look forward to receiving your revised manuscript.

Kind regards,

Luca Valera

Academic Editor

PLOS ONE

Journal Requirements:

Additional Editor Comments (if provided):

Please address the comments by reviewer 1

Reviewers' comments:

Reviewer's Responses to Questions

**Comments to the Author**

Reviewer #1: All comments have been addressed

Reviewer #2: All comments have been addressed

2. Is the manuscript technically sound, and do the data support the conclusions?

Reviewer #1: Partly

Reviewer #2: Yes

3. Has the statistical analysis been performed appropriately and rigorously?

Reviewer #1: No

Reviewer #2: Yes

4. Have the authors made all data underlying the findings in their manuscript fully available?

Reviewer #1: Yes

Reviewer #2: No

5. Is the manuscript presented in an intelligible fashion and written in standard English?

Reviewer #1: Yes

Reviewer #2: Yes

Reviewer #1: Thank you for study attending the comments that I have provided. Below are some observations that I believe could strengthen your manuscript.

Justification of Methods Selection:

Although the selection of the Rosenwax and Murtagh methods is well justified, a brief discussion on why alternative methods, such as the Lancet Commission approach, were not considered would further strengthen the methodological rationale.

Impact of Estimation Categories:

The authors opted for the “medium” category in the Rosenwax method and “high medium” in the Murtagh method. While reasonable, a more explicit discussion of how these choices may influence the results compared to the minimum or maximum categories would improve the manuscript.

Limited Discussion of Methodological Alternatives:

The authors emphasize the adaptability and robustness of their selected methods but could briefly discuss the limitations of other approaches (e.g., Gómez-Batiste or Howard et al.) to provide a more balanced perspective.

Reviewer #2: Thank you again for the opportunity to review this paper. I think the authors have addressed all reviewers comments and the methods and limitations of the research are now clearer in the manuscript.

I have no further comments.

**Do you want your identity to be public for this peer review?** For information about this choice, including consent withdrawal, please see our Privacy Policy

Reviewer #1: No

Reviewer #2: **Yes: ** Javiera Léniz

---

## [Author Response · Author response to Decision Letter 2]

3 Feb 2025

Reviewer #1:

1. Justification of Methods Selection: Although the selection of the Rosenwax and Murtagh methods is well justified, a brief discussion on why alternative methods, such as the Lancet Commission approach, were not considered would further strengthen the methodological rationale.

R: Dear Reviewer, we appreciate your valuable observation. We have included a paragraph explaining why the approach proposed by the Lancet Commission was not used:

<< We chose not to use the Lancet Commission approach, although we recognize its relevance, for two reasons: (1) the methods proposed by Rosenwax and Murtagh are better suited for analyzing hospital deaths and specific diagnoses, which are directly applicable to our population, whereas the Lancet Commission approach is more appropriate for regional or national-level estimations; (2) the methods proposed by Rosenwax and Murtagh methods ensure greater comparability with previous studies conducted in similar contexts and significantly contribute to the literature in low- and middle-income countries. >>

2. The authors opted for the “medium” category in the Rosenwax method and “high medium” in the Murtagh method. While reasonable, a more explicit discussion of how these choices may influence the results compared to the minimum or maximum categories would improve the manuscript.

R: Dear reviewer, thank you for your valuable feedback. We clarify the reasons behind selecting the "medium" category in the Rosenwax method and the "high-medium" category in the Murtagh method for this study. The "medium" category in the Rosenwax method allows for the inclusion of hospitalizations related to key diagnoses within the past 12 months, avoiding both the underestimation of patients who may require palliative care (as in the "minimum" category) and the overestimation of irrelevant cases (as in the "maximum" category). Similarly, the "high-medium" category in the Murtagh method incorporates emerging conditions and contributing diagnoses, ensuring more precise estimates aligned with previous studies, without resorting to excessive generalizations. However, we believe that including this level of detail in the manuscript might overwhelm the reader with additional information that could be unnecessary.

3. Limited Discussion of Methodological Alternatives: The authors emphasize the adaptability and robustness of their selected methods but could briefly discuss the limitations of other approaches (e.g., Gómez-Batiste or Howard et al.) to provide a more balanced perspective.

R: Dear reviewer, thank you for your comment. Regarding the limitations of other approaches, these have been addressed in the methodology section, where the selection of the methods proposed by Rosenwax and Murtagh is also justified. For this reason, we believe it is not necessary to reiterate them in the discussion.

---

## [Editor Report · Decision Letter 2]

17 Feb 2025

Estimation of the demand for palliative care in a hospital in southern Peru

PONE-D-24-33922R2

Dear Dr. Alvaro Taype-Rondan,

We’re pleased to inform you that your manuscript has been judged scientifically suitable for publication and will be formally accepted for publication once it meets all outstanding technical requirements.

Kind regards,

Luca Valera

Academic Editor

PLOS ONE
---

## [Editor Report · Acceptance letter]

PONE-D-24-33922R2

PLOS ONE

Dear Dr. Taype-Rondan,

I'm pleased to inform you that your manuscript has been deemed suitable for publication in PLOS ONE. Congratulations! Your manuscript is now being handed over to our production team.

Kind regards,

on behalf of

Dr. Luca Valera

Academic Editor

PLOS ONE